# LLM4GRN: Discovering Causal Gene Regulatory Networks with LLMs – Evaluation through Synthetic Data Generation

## Abstract

Gene regulatory networks (GRNs) represent the causal relationships between transcription factors (TFs) and target genes in single-cell RNA sequencing (scRNA-seq) data. Understanding these networks is crucial for uncovering disease mechanisms and identifying therapeutic targets. In this work, we investigate the potential of large language models (LLMs) for GRN discovery, leveraging their learned biological knowledge alone or in combination with traditional statistical methods. We employ a task-based evaluation strategy to address the challenge of unavailable ground truth causal graphs. Specifically, we use the GRNs suggested by LLMs to guide causal synthetic data generation and compare the resulting data against the original dataset. Our statistical and biological assessments show that LLMs can support statistical modeling and data synthesis for biological research.

## 1 Introduction

Single-cell RNA sequencing (scRNA-seq) is a cutting-edge technology that enables the collection of gene expression data from individual cells. This approach opens up new avenues for a wide range of scientific and clinical applications. One crucial application of scRNA-seq data is the reconstruction and analysis of gene regulatory networks (GRNs), which represent the interactions between genes. GRN analysis can deepen our understanding of disease mechanisms, identify key regulatory pathways, and provide a foundation for the development of interventional gene therapies and targeted drug discovery.

Statistical causal discovery algorithms (Mercatelli et al., 2020; Scheines et al., 1998; Zheng et al., 2018; Brouillard et al., 2020; Lippe et al., 2021; Roohani et al., 2024; Yu & Welch, 2022) can reveal potential causal links between TFs and their target gene. However, they often lack robustness and are prone to detecting spurious correlations, especially in high-dimensional, noisy single-cell data. Furthermore, many of these approaches rely heavily on prior knowledge from curated databases (e.g., TRANSFAC (Wingender et al., 1996), RegNetwork (Liu et al., 2015), ENCODE (de Souza, 2012), BioGRID (de Souza, 2012), and AnimalTFDB (Hu et al., 2019)), which frequently lack essential contextual information such as specific cell types or conditions, leading to inaccuracies in the inferred regulatory relationships (Zinati et al., 2024).

The recent advancements in and success of large language models (LLMs) have opened up new possibilities for their use in scientific discovery (Sheth et al., 2024; Lu et al., 2024; AI4Science & Quantum, 2023), including causal discovery (Kıcıman et al., 2023; Abdulaal et al., 2023; Kasetty et al., 2024; Vashishtha et al., 2023; AI4Science & Quantum, 2023; Abdulaal et al., 2023; Khatibi et al., 2024). Most of the above methods involve the refinement of the statistically inferred causal graph by LLM. However, LLMs excel at synthesizing vast amounts of heterogeneous knowledge, making them well-suited to tasks that require the integration of diverse datasets, such as constructing full causal graphs based on scientific literature (Sheth et al., 2024). In addition, LLMs have already been shown to perform various genomic data analysis tasks (Märtens et al.; Jin et al., 2024; Tang & Koo, 2023; Fang et al., 2024; Toufiq et al., 2023; Elsborg & Salvatore, 2023), including foundation models pre-trained for genomic tasks (Wang et al., 2024a; Cui et al., 2024).

Inspired by recent advancements, we harness LLMs for inferring gene regulatory networks (GRNs) from scRNA-seq data. Specifically, we utilize LLMs either to generate complete GRNs (causal

graphs) directly or to provide prior knowledge in the form of a list of potential transcription factors (TFs), which can then be integrated into traditional statistical causal discovery algorithms.

We use causal synthetic data generation as a downstream task to evaluate GRNs or priors suggested by LLMs in the absence of reliable ground-truth graphs. This method embeds gene regulatory networks into the scRNA-seq data generation process and has been shown to preserve biological plausibility better (Zinati et al., 2024). By comparing synthetic data to an oracle dataset, we assess the practical utility of the inferred causal graph. Our results highlight the potential of general-purpose LLMs for GRN inference, as evidenced by both statistical and biological metrics. The best performance is achieved by combining LLMs with statistical GRN inference, pointing to a promising direction for scRNA-seq data analysis.

**Our contributions**:

1. We show the effectiveness of using out of the box Large Language Models (LLMs) for inferring gene regulatory networks (GRNs), showcasing their ability to capture complex biological interactions.

2. Comprehensive performance evaluation based on synthetic data generation. Our methodology effectively addresses the challenge posed by the absence of ground-truth causal graphs, facilitating a more rigorous assessment of inference methods.

3. Extensive biological insight on the best-performing LLM for PBMC-All dataset.

## 2 RELATED WORKS

**LLMs and Causality.**   Gene regulatory network inference from scRNA-seq data traditionally relies on statistical causal discovery methods (Pratapa et al., 2020; Huynh-Thu et al., 2010; Moerman et al., 2019). However, causal discovery often require external knowledge in the form of interventions (Brouillard et al., 2020), expert input (Kleinegesse et al., 2022), or priors from curated databases (Zinati et al., 2024). Recent advances in large language models (LLMs) offer a promising solution, as LLMs excel at integrating diverse knowledge and providing contextual information (Wan et al., 2024; Ma, 2024). Many of the recent works leverage LLMs for causal discovery by utilizing metadata, such as variable names, to infer causal relationships (Kıcıman et al., 2023; Ban et al., 2023b; Vashishtha et al., 2023; Ban et al., 2023a). Further optimizations, including more advanced prompting strategies beyond pairwise variable comparisons, have been developed to enhance causal discovery (Vashishtha et al., 2023; Jiralerspong et al., 2024). Further (Sheth et al., 2024) explored the effectiveness of completing a partial causal graph across diverse domains. This work, however, considers LLM as an oracle to discover causal graphs for GRN inference.

**LLMs and Biology.**   Models based on transformer architecture and trained on DNA or RNA genomic sequences are effective at prediction and generation tasks (Zhang et al., 2024). However, the availability of generative Gene-LLMs is limited, they lack rich contextual information and are focused on specific tasks (Zhang et al., 2024). To overcome these limitations, foundation models are tailored for the genomic tasks (Wang et al., 2024a; Cui et al., 2024). However, models trained on the genetic data do not achieve the wide context of LLMs like GPT4 that inject information from the scientific literature and freely available genomic databases. LLMs have been used for the tasks such as enhancing gene perturbation generators (Märtens et al.), protein interaction prediction (Jin et al., 2024), gene selection (Toufiq et al., 2023), analyzing biomarkers (Elsborg & Salvatore, 2023) and cell annotation (Fang et al., 2024). To the best of our knowledge general purpose LLMs have not been used for gene regulatory network inference.

**Causal Synthetic Data Generation.**   Causally synthetic data generation is an approach that focuses on embedding true causal relationships within the generated data. Several approaches have been proposed in which the generator is guided by the causal acyclic graph (Cinquini et al., 2021; van Breugel et al., 2021; Wen et al., 2021; Wang et al., 2024b; Bandyopadhyay & Sarkar, 2023). We use a recently proposed causal GAN designed for scRNA-seq data generation, shown to produce more biologically plausible results (Zinati et al., 2024).

## 3 PRELIMINARIES

### 3.1 CAUSAL GRAPH

A directed acyclic graph (DAG) $\mathcal{G} = (\mathbf{V}, \mathcal{E})$ is composed of a set of variables/vertices $\mathbf{V}$ and a set of (directed) edges $\mathcal{E}$ between them such that no cycle is formed. Let $P$ be the probability distribution over the same set of variables $\mathbf{V}$. $\mathcal{G}$ and $P$ satisfy the Markov condition if every variable is conditionally independent of its non-descendants given its parents. Assuming the Markov condition, the joint distribution of variables $V_1, V_2, \ldots \in \mathbf{V}$ can be factorized as:

$$P[V_1, V_2, \ldots, V_d] = \prod_i P[V_i | Pa(V_i)] . \tag{1}$$

where $Pa(V_i)$ denotes the set of parents of $V_i$.

A bipartite acyclic graph $\mathcal{G} = (\mathbf{U}, \mathbf{V}, \mathcal{E})$ is an instance of a DAG whose vertices are divided into two subsets $\mathbf{U}$ and $\mathbf{V}$. Each edge $\mathcal{E}_i$ connects only two vertices from $\mathbf{U}$ and $\mathbf{V}$.

#### 3.1.1 GENE REGULATORY NETWORK (GRN)

A gene regulatory network (GRN) is a collection of molecular regulators that interact with each other and with other substances in the cell to control the gene expression levels of mRNA and proteins. GRNs describe the relationships between genes, transcription factors (TFs), RNA molecules, and other regulatory elements within a biological system, illustrating how genes are turned on or off and how their expression is modulated over time and in different conditions. GRN can be expressed as a directed acyclic graph (DAG) 3.1. In this work we are considering a bipartite DAG consisting of TFs and target genes.

### 3.2 GENERATIVE ADVERSARIAL NETWORK (GAN)

A Generative Adversarial Network (GAN) consists of two neural networks, the generator $G$ and the discriminator $D$, which are trained simultaneously in a zero-sum game. The generator $G$ maps a random noise vector $\mathbf{z}$ sampled from a prior distribution $p_{\mathbf{z}}(\mathbf{z})$ (typically Gaussian) to the data space, producing a synthetic sample $G(\mathbf{z})$. The discriminator $D$ maps an input $\mathbf{x}$ to a scalar value $D(\mathbf{x}) \in [0, 1]$, representing the probability that $\mathbf{x}$ is a real sample.

The training objective for GANs is formulated as a minimax game defined by the following function:

$$\min_G \max_D V(D, G) = \mathbb{E}_{\mathbf{x} \sim p_{\text{data}}(\mathbf{x})}[\log D(\mathbf{x})] + \mathbb{E}_{\mathbf{z} \sim p_{\mathbf{z}}(\mathbf{z})}[\log(1 - D(G(\mathbf{z})))]$$

where $p_{\text{data}}(\mathbf{x})$ is the distribution of real data, and $p_g(\mathbf{x})$ is the distribution induced by $G$. The discriminator tries to maximize the probability of distinguishing real data from $G$'s samples, while $G$ aims to minimize $\log(1 - D(G(\mathbf{z})))$ to generate realistic data. At equilibrium, the optimal generator replicates the true data distribution, making $p_g(\mathbf{x}) = p_{\text{data}}(\mathbf{x})$. The Wasserstein GAN (WGAN) introduces the Wasserstein distance as an alternative to the Kullback-Leibler or Jensen-Shannon divergences traditionally used in GANs. This distance metric measures the similarity between the distributions of real data and the generated data, leading to more stable training and mitigating issues like mode collapse (Arjovsky et al., 2017). WGAN with Gradient Penalty (WGAN-GP) (Gulrajani et al., 2017) further enhances this stability by replacing the original weight clipping technique of WGAN with a gradient penalty, addressing problems such as vanishing or exploding gradients and underutilization of model capacity.

#### 3.2.1 GROUNDGAN

Causal GAN (CGAN) proposed by Kocaoglu et al. (2018) is a variant of GAN that incorporate a causal directed acyclic graph (DAG) into the data generation process. GRouNdGAN (Zinati et al., 2024) builds on CGAN, integrating components such as a causal controller, target generators, a critic, a labeler, and an anti-labeler. The training process for GRouNdGAN occurs in two stages. First, the causal controller, which generates transcription factor (TF) expression values, is pre-trained as the generator in a WGAN-GP framework. In the second stage, the pre-trained causal controller's TF

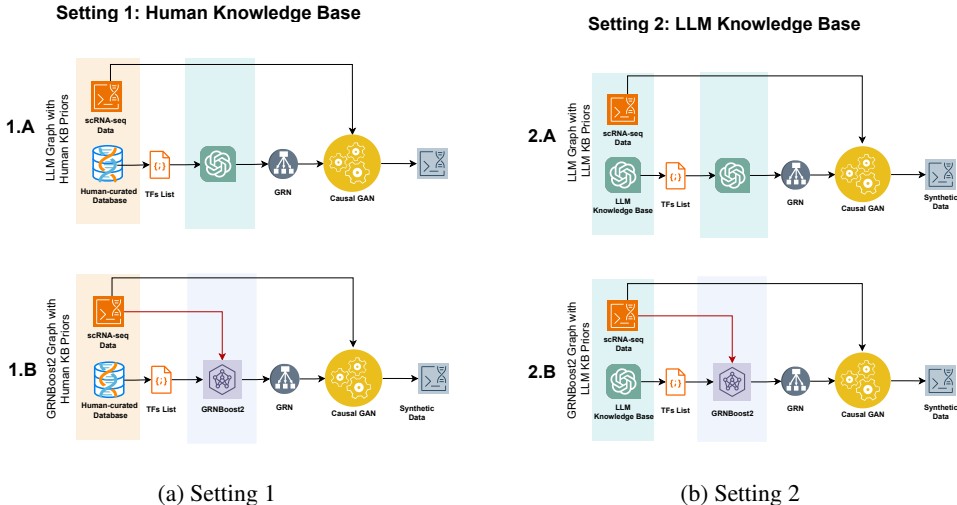

(a) Setting 1                                        (b) Setting 2

Figure 1: **Overview of LLM4GRN.** Setting **1.A** combines Human Knowledge Base (KB) with LLM. Setting **1.B** is the baseline setting that combines Human KB with GRNBoost2. Setting **2.A** is full LLM pipeline that combines LLM KB and LLM Inference. Setting **2.B** combines LLM KB with GRNBoost2 Inference.

expression values, along with randomly generated noise, are provided as input to target generators. These target generators, each an instance of WGAN-GP, produce the expressions of target genes while respecting the TF-gene relationships dictated by the input gene regulatory network (GRN). For a detailed explanation of GRouNdGAN's architecture and training, please refer to the original paper.

## 4   LLM4GRN

In this work, we propose a novel approach integrating Large Language Models (LLMs) for GRN inference. The overall goal is to leverage the potential of LLMs in capturing complex biological interactions and to assess their utility. Given the lack of ground-truth data, we consider causal synthetic data generation as a downstream task to perform biological and statistical evaluations. Our methodology involves two distinct experimental settings that leverage different knowledge about the potential transcription factors (TFs) to guide gene regulatory network (GRN)-informed data generation. In each of the settings, we introduce LLMs into the pipeline motivated by their ability to incorporate extensive contextual information.

- In the first setting ( Figure 1a, Setting **1.A**), we use LLM to infer the GRN graph by providing the LLM with a potential list of TF candidates sourced from a human-curated database.
- In the second setting ( Figure 1b, Setting **2.A**), we utilize the LLM as the knowledge base, incorporating it earlier in the pipeline to infer potential TFs and to deduce the GRN graph.

We compare with Setting **1.B** and **2.B** where the Human knowledge base and LLM knowledge base is used by a statistical causal inference approach, GRNBoost2, respectively.

### 4.1   GRN GRAPH

To model the regulatory relationships between transcription factors and target genes. In this framework, we maintain a knowledge base (KB) that contains comprehensive information regarding the regulatory interactions, specifically indicating which genes are target genes of specific transcription factors. KB takes as input a list of genes and outputs the corresponding target genes and their regulating transcription factors:

$$\text{KB} : \text{List of Genes} \rightarrow \{(\mathbf{T}_i, \mathbf{R}_j) \mid \mathbf{T}_i \in \mathbf{T}, \mathbf{R}_j \in \mathbf{R}\}$$

We represent the gene regulatory network as a *bipartite graph* $\mathcal{G} = (\mathbf{T}, \mathbf{R}, \mathcal{E})$, where $\mathbf{T}$ represents transcription factors (TFs), $\mathbf{R}$ contains target genes regulated by these TFs. The set $\mathcal{E}$ includes directed edges, where an edge $(\mathbf{T}_i, \mathbf{R}_j) \in \mathcal{E}$ indicates that transcription factor $\mathbf{T}_i$ regulates target gene $\mathbf{R}_j$.

## 4.2 SETTING 1: HUMAN KNOWLEDGE BASE

Given the ability of large language models (LLMs) to generate causal graphs from metadata (such as variable names) (Abdulaal et al., 2023), we establish the foundational components of gene regulatory networks (GRNs) using a human knowledge base, denoted as $\mathrm{KB}^H$. Let $\mathbf{T}^H$ represent the set of transcription factors identified through the human-curated database, and let $\mathbf{R}^H$ denote the set of target genes regulated by these factors. Total set of genes are defined by $G$. The directed edges in the graph are represented as $\mathcal{E}$, where an edge $(\mathbf{T}_i^H, \mathbf{R}_j^H) \in \mathcal{E}$ signifies that transcription factor $\mathbf{T}_i^H$ regulates target gene $\mathbf{R}_j^H$.

In Setting **1A**, ( Figure 1a), the LLM is employed to establish causal relationships between TFs and target genes, utilizing a list of TFs transcription factor candidates sourced from a human-curated database. We model the LLM as a function $\mathcal{F}_{\mathrm{LLM}}$ that, given a set of metadata $\mathcal{M}$, produces a bipartite graph $\mathcal{G} = (\mathbf{T}, \mathbf{R}, \mathcal{E})$, expressed as:

$$\mathcal{F}_{\mathrm{LLM}}(\mathcal{M}, \mathbf{T}^H, \mathbf{R}^H) = \mathcal{G} \tag{2}$$

The input metadata $\mathcal{M}$ encompasses gene names, transcription factors (TFs), single-cell RNA sequencing (scRNA-seq) data, and relevant biological context (such as species or experimental conditions) sourced from relevant literatures about the dataset.

It is important to highlight that, unlike traditional inference methods like GRNBoost2, the LLM-based GRN inference approach in this work does not rely on observational data, ensuring that individuals' privacy in the dataset remains uncompromised. By integrating metadata from scRNA-seq datasets, the LLM can construct GRNs that are tailored specifically for the biological context of the data, potentially capturing nuances that statistical methods cannot. Detailed descriptions of the various prompting strategies employed can be found in the Appendix A.2.

## 4.3 SETTING 2: LLM KNOWLEDGE BASE

In Setting **2A**, ( Figure 1b), we utilize LLM knowledge base, denoted as $\mathrm{KB}^{\mathrm{LLM}}$, to establish partition between transcription factors and target genes. In this context, the LLM is tasked with extracting relevant information directly from its knowledge base, which includes extensive biological data and relationships derived from various sources.

$$\mathcal{F}_{\mathrm{LLM}}(\mathcal{M}) = \left( \mathbf{T}^{\mathrm{LLM}}, \mathbf{R}^{\mathrm{LLM}} \right) \tag{3}$$

Here, $\mathbf{T}^{\mathrm{LLM}}$ denotes the set of transcription factors identified from the LLM knowledge base, and $\mathbf{R}^{\mathrm{LLM}}$ represents the corresponding target genes. The input metadata $\mathcal{M}$ includes gene names, transcription factors (TFs), and relevant biological context, leveraging the extensive knowledge embedded in the LLM.

## 4.4 GRN-INDUCED CAUSAL SYNTHETIC DATA GENERATION

We use the GRNs obtained by either of the settings, to perform synthetic causal data generation. These GRNs are fed into a causal GAN algorithm, specifically GRouNdGAN (Zinati et al., 2024), which uses them to generate synthetic datasets that correspond to each GRN structure in a two-stage approach.

## 5 RESULTS

We evaluate and compare the resulting synthetic datasets based on a range of statistical and biological metrics, allowing us to assess the quality and biological relevance of the data produced in each setting. Additionally, we analyze the TFs list generated by the LLM to gain insights, comparing them to priors from human curated database.

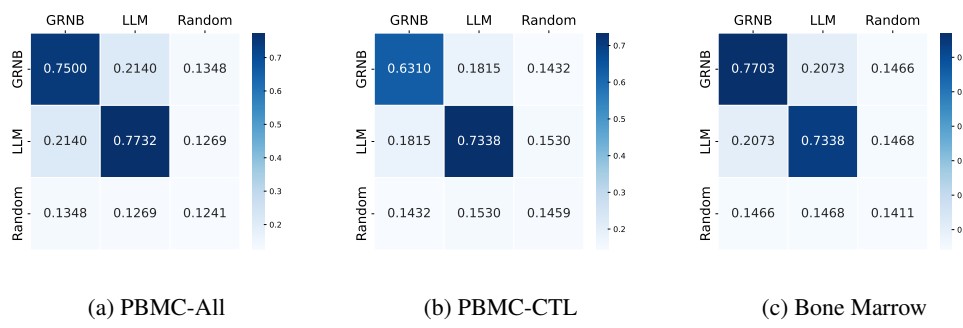

Figure 2: Overlap between GRNs proposed by different methods.

**Experimental Setup.** We generated synthetic data using datasets and protocols from (Zinati et al., 2024). Our preprocessing focused on creating train, test, and validation sets while maintaining 1,000 genes across all datasets (see AppendixA.1 for details). For each setting, we construct three different GRNs: one derived directly from the LLM, one generated by a causal discovery algorithm that incorporates the prior information (the dataset), and a third, randomly generated graph (based on TF list extracted from Human or LLM KB). For the GRN inference of LLM, we prompted with contextual knowledge. We used the state-of-art pretrained GPT-4 model (Achiam et al., 2023). Given GPT-4 strong performance for causal discovery (Abdulaal et al., 2023) across different domains including genomics, we prompted (see Appendix A.2) in zero-shot fashion. We also compare open-source model, Llama-3.1-70B in Setting 2.

**Evaluation.** We employed four statistical metrics: Cosine and Euclidean distances to measure the differences between centroids, maximum mean discrepancy (MMD) to assess the proximity of high-dimensional distributions without centroid creation, and Random Forest Area Under the Receiver Operating Characteristic (RF-AUROC) to determine the distinguishability of real and synthetic cells. We evaluate biological plausibility of the datasets by performing gene expression analyis and cell type annotation task. For more details, refer to Appendices A and B.

**Datasets.** For the direct comparison of the different settings, we focus on the dataset and genomic database information by (Zinati et al., 2024). Specifically, PBMC-All, PBMC-CTL and BoneMarrow data sets (details in Appendix A.1.1). Unlike Zinati et al. (2024), whose goal is to generate causal synthetic scRNA datasets, our objective is to evaluate various GRN graphs. We employ causal synthetic data generation as a means to address the lack of reliable ground truth.

## 5.1 COMPARISION AGAINST BASELINE

**Evaluation of LLM graphs.** For Setting 1, in the absence of ground truth for the GRN, we compare the overlap between the graph proposed by GRNBoost2, a statistical method, and the graph hypothesized by the LLM. While GRNBoost2 is not the definitive ground truth, it serves as a useful reference point. Analyzing this overlap allows us to assess how the hypothesized connections align with established statistical methods and examine how these differences impact downstream synthetic data generation metrics. As a baseline, we also compared against a randomly generated causal graph. Additionally, we test the consistency of the LLM's performance across different random seeds by measuring the overlap between graphs produced from multiple seeds. This helps us evaluate the robustness and stability of the LLM-generated hypotheses. We plot the overlaps for all of the datasets in Figure 2. Our two main observations are that (1) the LLM-derived GRN demonstrates greater robustness compared to the GRNBoost2 (GRNB) GRN, particularly in terms of higher certainty; and (2) the overlap between the LLM-generated GRN and the random GRN is smaller than between LLM GRN and GRNBoost2 GRN, suggesting, that the LLM could be generating meaningful graphs.

**Evaluation of LLM KB.** In Setting 2, we introduced the LLM to filter TF and target genes from a list of genes. Similar to calculating overlaps in GRN evaluation, we also compute the overlap of TF produced by both approaches. From Table 5 (Appendix C), we observe there exists around just about half an overlap between the two knowledge bases. Interestingly we observe less than 50% overlap

between KB$^H$ and KB$^{LLM}$ for PBMC-CTL while a much higher overlap for the PBMC-ALL dataset. We also observed that LLM proposed a higher number of TFs, over 2 times in the case of the Bone Marrow dataset. As stated in the Table 6 (Appendix C), would change the density of the graphs potentially affecting the downstream tasks.

## 5.2 STATISTICAL EVALUATION OF GRN INFERENCE METHODS ON SYNTHETIC DATA

In Table 1, we present the statistical metrics results for the three datasets. Metrics are computed between a synthetic dataset of 1000 cells and a held-out test set of 1000 real cells for PBMC-ALL and PBMC-CTL. For BoneMarrow, 500 samples were used for synthetic and held-out test set. In GRouNdGAN's imposed GRN, each gene is regulated by 10 transcription factors (TFs). Lower values (↓) indicate better performance for all metrics, with the first two metrics representing the distance between the centroids of the real and synthetic cells. The "control" metrics are based on the real training dataset. The best performance values (excluding control and Stage1 which is a non-causal baseline) are highlighted in bold. Evaluations were carried out on 4 synthetic datasets, with experiments repeated using 2 cross-validation seeds.

**Setting 1 Comparison.** The Setting 1 (Table 1) result shows the LLM (GPT-4)-inferred GRN achieves the best performance across all metrics. For example, for PBMC-ALL, the LLM model shows the lowest Cosine distance of 0.00024 and Euclidean distance of 89, indicating that the synthetic data generated is most similar to the real data in terms of overall structure. Its lower MMD of 0.0072 compared to the GRNBoost2 model suggests it effectively captures subtle gene expression distributions. Additionally, the LLM model performs well on the Random Forest metric of 0.63, which measures binary classification accuracy in distinguishing real from synthetic data. Unsurprisingly, the Random Graph method performs the worst across all metrics, particularly with a high MMD of 0.0166 and the largest Euclidean distance of 121, highlighting the importance of informed GRN inference methods. In contrast, for the PBMC-CTL and BoneMarrow, the GRNBoost2 graph outperform the LLM (GPT-4) in this setting.

**Setting 2 Comparison.** In Setting 2 (Table 1), where we incorporated the LLM knowledge base, the GRNBoost2 method—combining LLM-proposed transcription factor (TF) lists with GRN inference—outperforms the fully LLM-based GRN approach (i.e., where the LLM proposes both the TF list and the connections between TFs and genes) across all datasets. GRNBoost2 also achieves the best overall performance for all datasets. For instance, in the PBMC-ALL dataset, it achieves a Euclidean distance of 83, improving from 121 in Setting 1, and an RF AUROC of 0.59, compared to 0.73 in Setting 1. Overall, GRNBoost2 shows improvement over its performance in Setting 1, delivering better results across all metrics for both PBMC-ALL and PBMC-CTL. Notably, it surpasses the top results from Setting 1, where the LLM achieved a Cosine distance of 0.00024 and a Euclidean distance of 89 for PBMC-ALL, while GRNBoost2 reached a Cosine distance of 0.00025 and a Euclidean distance of 65 for PBMC-CTL. These findings suggest that integrating LLM-derived priors with GRNBoost2's statistical inference yields a more accurate and robust representation of gene regulatory networks (GRNs). In contrast, BoneMarrow performed best in Setting 1 but remains competitive in Setting 2 when the LLM knowledge base is combined with GRNBoost2.

**Setting 2 Comparison using Llama-3.1-70B as Knowledge Base.** Table 2 presents the results of Setting 2 for Llama-3.1-70B Knowledge Base. The findings indicate that Llama, when queried for knowledge, achieves the best overall performance in combination with GRNBoost2, outperforming GPT-4's knowledge base (Table 1) across all metrics. These results are promising, suggesting that integrating LLM knowledge bases with causal inference approaches like GRNBoost2 offers a highly promising direction for scRNA causal synthetic data generation.

## 5.3 BIOLOGICAL PLAUSIBILITY OF CAUSAL SYNTHETIC DATA

We perform gene expression analysis and cell-type annotation tasks on the best statistically best-performing datasets. Full analysis can be found in Appendix D( subsection C.3).

| | | Cosine distance ↓ | Euclidean distance ↓ | MMD ↓ | RF AUROC ↓ |
|---|---|---|---|---|---|
| **PBMC-ALL** | *Baseline* | | | | |
| | Control | $0.00029_{\pm 0.00008}$ | $100_{\pm 16}$ | $0.0051_{\pm 0.001}$ | $0.49_{\pm 0.017}$ |
| | Stage 1 | $0.00036_{\pm 0.00009}$ | $107_{\pm 15}$ | $0.0057_{\pm 0.005}$ | $0.55_{\pm 0.021}$ |
| | *Setting 1 Human-KB* | | | | |
| | GPT-4 | $\underline{0.00024}_{\pm 0.00004}$ | $\underline{89}_{\pm 7}$ | $\underline{0.0072}_{\pm 0.001}$ | $\underline{0.63}_{\pm 0.043}$ |
| | GRNBoost2 | $0.00047_{\pm 0.00021}$ | $121_{\pm 25}$ | $0.0139_{\pm 0.006}$ | $0.73_{\pm 0.050}$ |
| | Random | $0.00045_{\pm 0.00013}$ | $121_{\pm 22}$ | $0.0166_{\pm 0.003}$ | $0.85_{\pm 0.019}$ |
| | *Setting 2 GPT-4-KB* | | | | |
| | GPT-4 | $0.00026_{\pm 0.00009}$ | $90_{\pm 13}$ | $0.0206_{\pm 0.001}$ | $0.86_{\pm 0.018}$ |
| | GRNBoost2 | $\mathbf{0.00023}_{\pm 0.00008}$ | $\mathbf{83}_{\pm 17}$ | $\mathbf{0.0069}_{\pm 0.001}$ | $\mathbf{0.59}_{\pm 0.028}$ |
| | Random | $0.00026_{\pm 0.00011}$ | $92_{\pm 14}$ | $0.0226_{\pm 0.002}$ | $0.87_{\pm 0.018}$ |
| **PBMC-CTL** | *Baseline* | | | | |
| | Control | $0.00020_{\pm 0.00005}$ | $57_{\pm 7}$ | $0.0045_{\pm 0.000}$ | $0.54_{\pm 0.007}$ |
| | Stage 1 | $0.00023_{\pm 0.00003}$ | $63_{\pm 4}$ | $0.0049_{\pm 0.000}$ | $0.57_{\pm 0.030}$ |
| | *Setting 1 Human-KB* | | | | |
| | GPT-4 | $0.00265_{\pm 0.00253}$ | $176_{\pm 115}$ | $0.0238_{\pm 0.020}$ | $0.79_{\pm 0.206}$ |
| | GRNBoost2 | $\underline{0.00025}_{\pm 0.00004}$ | $\underline{65}_{\pm 6}$ | $\underline{0.0053}_{\pm 0.000}$ | $\mathbf{0.59}_{\pm 0.028}$ |
| | Random | $0.00027_{\pm 0.0001}$ | $66_{\pm 7}$ | $0.0085_{\pm 0.001}$ | $0.75_{\pm 0.021}$ |
| | *Setting 2 GPT-4-KB* | | | | |
| | GPT-4 | $0.00028_{\pm 0.00009}$ | $67_{\pm 11}$ | $0.0067_{\pm 0.001}$ | $0.70_{\pm 0.025}$ |
| | GRNBoost2 | $\mathbf{0.00020}_{\pm 0.00004}$ | $\mathbf{57}_{\pm 6}$ | $\mathbf{0.0049}_{\pm 0.000}$ | $\mathbf{0.59}_{\pm 0.019}$ |
| | Random | $0.00022_{\pm 0.00005}$ | $59_{\pm 7}$ | $0.0080_{\pm 0.000}$ | $0.76_{\pm 0.023}$ |
| **Bone Marrow** | *Baseline* | | | | |
| | Control | $0.00205_{\pm 0.00018}$ | $80_{\pm 4}$ | $0.0109_{\pm 0.001}$ | $0.60_{\pm 0.037}$ |
| | Stage 1 | $0.00320_{\pm 0.00115}$ | $101_{\pm 16}$ | $0.0197_{\pm 0.007}$ | $0.66_{\pm 0.037}$ |
| | *Setting 1 Human-KB* | | | | |
| | GPT-4 | $0.00238_{\pm 0.00052}$ | $86_{\pm 9}$ | $0.0124_{\pm 0.001}$ | $0.70_{\pm 0.080}$ |
| | GRNBoost2 | $\mathbf{0.00190}_{\pm 0.00023}$ | $\mathbf{77}_{\pm 5}$ | $\mathbf{0.0118}_{\pm 0.001}$ | $\mathbf{0.64}_{\pm 0.023}$ |
| | Random | $0.00219_{\pm 0.00030}$ | $82_{\pm 6}$ | $0.0150_{\pm 0.003}$ | $0.75_{\pm 0.041}$ |
| | *Setting 2 GPT-4-KB* | | | | |
| | GPT-4 | $0.00217_{\pm 0.00050}$ | $82_{\pm 10}$ | $0.0137_{\pm 0.003}$ | $0.72_{\pm 0.044}$ |
| | GRNBoost2 | $\underline{0.00193}_{\pm 0.00023}$ | $\underline{78}_{\pm 4}$ | $\underline{0.0119}_{\pm 0.001}$ | $\mathbf{0.64}_{\pm 0.033}$ |
| | Random | $0.00297_{\pm 0.00080}$ | $96_{\pm 13}$ | $0.0172_{\pm 0.004}$ | $0.79_{\pm 0.046}$ |

Table 1: **Performance of Different GRN Inference Methods in Simulating Realistic scRNA-seq Data across all 3 datasets.** The best value for each dataset is presented in **boldface** and the best value in each setting is underline. The lower (↓) the better for all metrics.

| | Cosine distance ↓ | Euclidean distance ↓ | MMD ↓ | RF AUROC ↓ |
|---|---|---|---|---|
| *Setting 2 Llama-KB* | | | | |
| Llama-3.1 | $0.00032_{\pm 0.00005}$ | $100_{\pm 8}$ | $0.0105_{\pm 0.0002}$ | $0.79_{\pm 0.02}$ |
| GRNBoost2 | $\mathbf{0.00016}_{\pm 0.00001}$ | $\mathbf{74}_{\pm 2}$ | $\mathbf{0.0064}_{\pm 0.0006}$ | $\mathbf{0.59}_{\pm 0.02}$ |
| Random | $0.00026_{\pm 0.00001}$ | $89_{\pm 1}$ | $0.0143_{\pm 0.0006}$ | $0.89_{\pm 0.01}$ |

Table 2: **Comparison using LLM (Llama-3.1-70B) as Knowledge Base - performance GRN Inference Methods in Simulating Realistic scRNA-seq Data for PBMC-ALL dataset.** The best value is presented in **boldface**. The lower (↓) the better for all metrics.

### 5.3.1 GENE EXPRESSION ANALYSIS

**General Performance of Llama-KB GRNBoost2 in Cell-Specific Expression Profiles**   Although Llama-KB GRNBoost2 synthetic data Figure 3b introduces some noise, it generally improves cell-

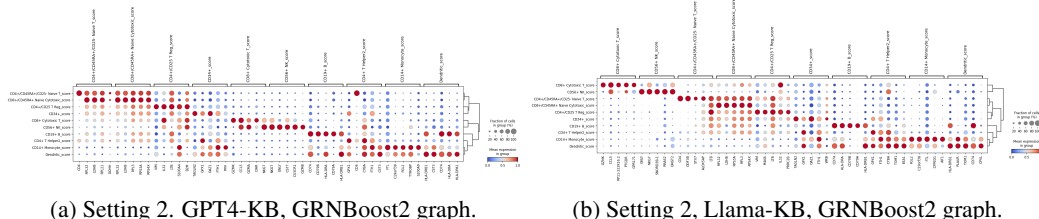

(a) Setting 2. GPT4-KB, GRNBoost2 graph.   (b) Setting 2, Llama-KB, GRNBoost2 graph.

Figure 3: Gene expression analysis.GPT4-KB GRNBoost2 and Llama-KB, GRNBoost2 datasets.

specific expression profiles, which could be further optimized for even cleaner cell type differentiation. A key observation was that when mean expression was higher in specific cell types, it tended to be lower in others, with fewer than 30% of cells displaying similar expression fractions. In contrast, when expression was noisy across multiple cell types, both mean expression and cell fraction tended to be similar across these groups.

**Human-KB GRNBoost2 Dataset Performance and Noise Patterns** In comparison, the GPT4-KB GRNBoost2 dataset Figure 3a exhibited more noise than the Llama-KB dataset Figure 3b, with multiple markers expressed across various cell types at comparable mean expression levels and with higher cell fractions. For the Human-KB GRNBoost2 dataset, noise was primarily observed in a few cell types, including CD4+/CD45A+/CD25– Naïve T cells and CD8+/CD45RA+ Naïve cytotoxic T cells. Some markers from other cell types also showed similar expression across multiple cell types. These noisy patterns were not observed in the Stage 1 or random datasets and were less pronounced in the Llama-KB and GPT4-KB models, though Llama-KB GRNBoost2 outperformed GPT4-KB GRNBoost2 in reducing noise for these specific cell types.

### 5.3.2 CELL TYPE ANNOTATION

**Variations in Cell Type Proportions Across Datasets** A notable observation is that the cell type proportions in each generated dataset differ significantly from those in the original dataset, indicating a noisy overall expression profile that can alter the distribution of cell types. In the Llama-KB GRNBoost2 dataset ( Figure 9b), CD8+/CD45RA+ Naïve Cytotoxic T cells were the most abundant at 34.1%, followed closely by CD56+ NK cells, CD8+ Cytotoxic T cells, and CD4+/CD25+ T regulatory cells. This trend was similarly observed GPT4-KB GRNBoost2 ( Figure 9a) and other datasets, albeit with slightly varying percentages.

**Overall Performance of Llama-KB GRNBoost2** We observe that CD4+/CD45RA+/CD25– Naïve T Cells and CD8+/CD45RA+ Naïve Cytotoxic T Cells exhibited noisy expression patterns consistently across all datasets. The original dataset performed better for CD8+/CD45RA+ Cytotoxic T Cells, showing lower noise in their expression compared to the Llama-KB GRNBoost2 ( Figure 9b) dataset. Among the various models, the Llama-KB GRNBoost2 model emerged as the best performer, providing clearer segregation of cell types and reduced noise, particularly for CD4+/CD45RA+/CD25– Naïve T Cells and dendritic cells. In contrast, the GPT4-KB GRNBoost2 ( Figure 9a) model exhibited more noise than Llama-KB GRNBoost2, with similar markers expressed across various cell types at comparable mean expression levels.

### 5.4 DISCUSSION AND LIMITATIONS

We observe promising results in using LLM for GRN discovery, especially on the PBMC dataset. The hybrid approach incorporating LLM TFs and GRNBoost2 causal discovery yield the best overall performance (for PBMC-ALL and PBMC-CTL data sets). Surprisingly, a smaller open-source model Llama has shown the best result in this setting. This result is surprising, however, although further evaluation is needed, new LLM models might perform as good or better than state-of-the-art GPT for specific tasks Valero-Lara et al. (2023). However, Llama performs worse on the more challenging, GRN construction task. The lower performance of LLMs on CTL and BoneMarrow data sets is not surprising. Recent studies suggest that genomic LLMs under-perform on cell-specific data (Tang & Koo, 2023). In addition, human transcription factors outnumber mice transcriptomic information

in the databases (Members & Partners, 2024) making the information on mice genes scarcer in the LLMs training data.

Biological plausibility analysis reveals that the Setting 2, Llama-KB dataset significantly improves cell type differentiation and reduces noise compared to other generated datasets, particularly for CD4+/CD45RA+/CD25– Naïve T cells and dendritic cells, which indicates better performance of the model due to its ability to handle large contexts (Xiong et al., 2023).

In contrast, while the Setting 2, GPT4-KB model exhibited higher noise and less specificity, the Setting 1, GRNBoost2 model also showed noisy patterns, especially in Naïve T and cytotoxic T cells, emphasizing the need for models that clearly delineate cell type-specific expression. Notably, discrepancies in cell type proportions across generated datasets, particularly the inflated presence of CD8+/CD45RA+ Naïve Cytotoxic T cells, raise concerns about the biological relevance of these models. As reported previously, large-scale single-cell studies tend to be more noisy which can lead to sub-optimal biological inferences (Kavran & Clauset, 2021). Therefore, further refinements are essential across all approaches to enhance specificity and reliability in representing true cellular compositions, which is crucial for accurate downstream biological analyses and interpretations. Finally, the Human-KB LLM model exhibited highly noisy or non-specific expression profiles, particularly for CD4+/CD45A+/CD25– Naïve T cells, CD8+/CD45RA+ Naïve cytotoxic T cells, CD4+ T helper cells, CD4+/CD25+ T regulatory cells, and CD4+/CD45RO+ memory cells. For other cell types, the model either failed to express most top markers or showed high cell fractions with reduced mean expression across different cell types.

One of the limitations of the study is the unifying constraints that are imposed on the GRN discovery due to the parametric requirements of GRouNdGan (Zinati et al., 2024). Namely, all GRN graphs are set to be bipartite graphs with the same number of TFs and same number of target genes for each TF. These constraints restrict the diversity of the generated graphs, resulting in fairly similar performance metrics among different GRN discovery approaches. In addition, multi-layer graphs with transcription factors, cofactors, and target genes are more realistic and can better reflect biological complexity (Karlebach & Shamir, 2008).

Despite the promising results, we note, that LLMs should be used cautiously in any high-stakes decision-making. The models are prone to reporting false information with confidence (Ji et al., 2023; Farquhar et al., 2024). In addition, they are susceptible to biases in the training data. One of the most common sources of bias in machine learning is the under-representation of minority populations in the training data. Transcription factor databases and literature often lack balanced representation across ethnicity, ancestry, gender, and age groups. The vast majority of genetic data and studies are based on individuals of European ancestry (Sirugo et al., 2019; Bentley et al., 2017). The underrepresentation of non-European ancestry populations in genomic databases can obscure gene-disease associations that are rare in European groups, leading to treatments that may not be effective for these communities (Landry et al., 2018; Tawfik et al., 2023; Barral-Arca et al., 2019). Age groups, specifically children and elderly, females, low-income populations and individuals from remote rural areas are also underrepresented in clinical trials and data, making them less likely to benefit from the achievements of precision medicine (Mosenifar, 2007; Davis et al., 2019; Steinberg et al., 2021). LLMs are shown to exacerbate some existing health disparities (Pfohl et al., 2024), and are likely to be influenced by the under-representation bias in the genomic data.

## 6 CONCLUSION

In conclusion, our analysis highlights the potential of using Large Language Models (LLMs) for gene regulatory network (GRN) inference, as supported by both statistical and biological performance metrics. Notably, the best performing approach combines LLM-derived TF priors with GRNBoost2 for statistical inference. In addition, we see good potential for using open-source models such as Llama for GRNs, and we aim to explore their utility further, either directly or with fine-tuning. Our biological results suggest that additional refinement is needed. For this we foresee developing GRNs tailored to individual cell types, as evidence indicates that GRNs are often cell-type-specific.

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

## A EXPERIMENTAL SETUP

### A.1 DATA PREPARATION AND MODIFICATIONS

In synthetic data generation, we use the same data sets and follow the protocol described by Zinati et al. Zinati et al. (2024).

We downloaded the three datasets using the link provided at `https://emad-combine-lab.github.io/GRouNdGAN/tutorial.html#demo-datasets`.

To ensure consistent results across different runs, we modified the code in two key ways: (1) we seeded the randomness in the 'main.py' file. While the GRNBoost2 algorithm is already seeded by default, ensuring consistent output, the rest of the code was not. Given that we are testing different GRN graphs, it was critical to control for randomness to prevent it from influencing our results. (2) We addressed additional bugs to ensure smooth execution, such as handling sparse data loading and ensuring that the code was properly assigned to the correct device for training. All modifications have been incorporated into our forked version of the repository, available upon publication. Some of the bugs we identified were also reported and fixed by the original authors.

After downloading the raw data, we preprocessed it to create separate train, test, and validation files. Following the paper's recommendations, we used a test set of 1,000 samples for the PBMC and CTL datasets, and 500 samples for the BoneMarrow dataset. For validation, we set aside 1,000 samples each for PBMC and CTL, and 500 for BoneMarrow. The subsets were disjoint. We refer to these preprocessed datasets as "Real Data" throughout the paper. The preprocessed data will be made available upon publication.

During preprocessing, we observed that the BoneMarrow dataset had two genes absent in the training data, as they were not expressed in any cells. This issue did not occur in the full dataset, where genes expressed in fewer than three cells and cells expressing fewer than 10 genes had been filtered out. However, after splitting the data into train, test, and validation sets, some genes in specific splits were not expressed at all.

To address this, we modified the code to filter out genes expressed in fewer than one cell for each data split. This reduced the number of genes in the BoneMarrow dataset from 1,000 to 910. In the PBMC dataset, we found that 493 genes were not expressed in the test set, requiring extensive filtering.

To resolve this, we revised our approach. Instead of applying the original threshold, which discarded genes expressed in fewer than three cells, we increased this threshold to 680 cells for the PBMC dataset, equivalent to 0.01% of the full dataset. This change allowed us to retain the full 1,000 highly variable genes. However, the resulting set of 1,000 genes may differ slightly from those identified with the initial criteria, given that the full PBMC dataset contains over 32,738 genes, while BoneMarrow contains 3,451 genes.

This revised strategy allowed us to retain the full 1,000 genes for the PBMC and CTL datasets but not for BoneMarrow. To address this, we further adjusted the strategy by filtering out genes expressed in fewer than the number of cells in the test set for each dataset (1,000 for PBMC and CTL, and 500 for BoneMarrow). This approach successfully retained 1,000 genes across all datasets.

After preprocessing, we followed the rest of the process as detailed in the tutorial provided by the authors.

### A.1.1 DATA SETS

- Human peripheral blood mononuclear cell (PBMC-Al). 68579 samples corresponding to 11 cell types

- Human peripheral blood mononuclear CD8+ Cytotoxic T-cells (PBMC-CTL). 20773 samples from the most common cell type in PBMC-All

- Mouse bone marrow Hematopoietic stem cells lineage differentiation (BoneMarrow). 2730 cells.

| Dataset | # TFs | # Targets | # Genes | # Possible Edges | # Imposed Edges | GRN density Edges |
|---------|-------|-----------|---------|------------------|-----------------|-------------------|
| PBMC | 75 | 925 | 1000 | 69375 | 9250 | 0.1333 |
| CTL | 65 | 935 | 1000 | 60775 | 9350 | 0.1538 |
| BoneMarrow | 68 | 932 | 1000 | 63376 | 9320 | 0.1471 |

Table 3: The density of the causal graphs.

## A.2 LLM PROMPTING STRATEGIES

In this study, we utilize prompting techniques to guide the behavior of a Large Language Model (LLM) for gene regulatory network (GRN) inference and other downstream analyses. The model employed is based on a pretrained large language model, specifically GPT-4, which has not been fine-tuned for this task. Instead, we rely on advanced prompting strategies, including Chain of Thought (CoT) reasoning and context provision, to enhance the performance of the LLM in generating biologically plausible results.

### A.2.1 CHAIN OF THOUGHT (COT) PROMPTING

Chain of Thought (CoT) prompting is a method that encourages the LLM to reason through intermediate steps, producing a more transparent and logical progression towards its final answer. By guiding the model to provide step-by-step explanations before arriving at a conclusion, we aim to improve the interpretability and accuracy of its predictions. CoT prompting is particularly valuable for tasks that require complex reasoning, such as GRN inference, where multiple factors, such as transcription factor interactions and gene expression patterns, must be considered.

### A.2.2 CONTEXT PROVISION

To further enhance the performance of the LLM, we utilize context provision by supplying the model with relevant background information prior to prompting. This technique is especially important when the task requires domain-specific knowledge, such as GRN inference from single-cell RNA sequencing (scRNA-seq) data. By embedding relevant context in the prompt, we can better align the model's responses with the biological characteristics of the data being analyzed. Specifically, we provide excerpts from the original articles where the analyzed data sets were introduced Paul et al. (2015); Zheng et al. (2017).

> We need to find transcriptomic factors related to CONTEXT. What are the transcriptomic factors genes related to GENE-X out of LIST-OF-TFs. Which of these 10 TFs have a causal relationship with GENE-X gene? Do not include any genes beyond these. Give potential candidates. Think step by step and return the answer in the format <Answer> [first suggestion, second suggestion, third suggestion, fourth suggestion and so on....] </Answer>. You have to return the 10 potential TFs that are related to the give gene only, otherwise your answer will be disqualified.

### A.2.3 EXTRACTING POTENTIAL TFS FROM THE GENE LIST

we provide the LLM with a curated list of genes, accompanied by relevant contextual information such as biological function, tissue type, and experimental conditions derived from scRNA-seq datasets. The prompt explicitly requests the LLM to identify and propose TFs that are known to regulate the expression of each gene within the list. In our approach, we start with a total of 1000 genes and sequentially query the Large Language Model (LLM) about subsets of 20 genes at a time to extract potential transcription factors (TFs). For each initial query, the LLM identifies and proposes TFs that may regulate the selected genes, leveraging its extensive contextual knowledge of gene regulatory mechanisms. In subsequent prompts, we maintain a 50% overlap with the previous set of genes, ensuring that 10 of the genes are revisited while introducing 10 new genes. This iterative process allows us to refine the extraction of TFs, incorporating insights from the previous queries while continuously expanding our understanding of the regulatory landscape. By doing so, we aim

to capture a comprehensive set of potential TFs that interact with the entire gene pool, facilitating a more robust inference of the Gene Regulatory Network (GRN).

> We need to find transcriptomic factors related to CONTEXT out of given genes. What are the transcriptomic factors genes out of LIST-OF-TFs. Which of these can be transcriptomic factors? Do not include any TFs beyond these. Give potential candidates. Return the answer in the format <Answer> [first suggestion, second suggestion, third suggestion, fourth suggestion and so on....] </Answer>. Describe the reasoning first and then return answer in requested format with the potential TFs.

## B  METRICS

### B.1  STATISTICAL EVALUATION METRICS

**Euclidean Distance:**  To compute the Euclidean distance between the centroids of the real and simulated cells, we first calculate the centroid by finding the mean along the gene axis across all simulated and real cells. The Euclidean distance $d(r, s)$ is then given by:

$$d(r, s) = \|\mu(R) - \mu(S)\|_2$$

Where $R$ amd $S$ are matrix of real and simulated cells, with elements $R_{i,j}$ and $S_{i,j}$ where $i$ indexes the cells and $j$ indexes the genes. $\mu(R)$ and $\mu S$ is the mean vector of the real and synthetic cells along the gene axis (columns) respectively.

$$\mu(R) = \left( \frac{1}{n} \sum_{i=1}^{n} R_{i,j} \right)_{j=1,\ldots,m} \quad \text{and } \mu(S) = \left( \frac{1}{n} \sum_{i=1}^{n} S_{i,j} \right)_{j=1,\ldots,m}$$

**Cosine Distance:**  This computes the cosine distance between the centroids of the real and simulated cells. The centroid is obtained by calculating the mean along the gene axis across all simulated and real cells.

$$d_{\cos}(r, s) = 1 - \frac{\mu(R) \cdot \mu(S)}{\|\mu(R)\|_2 \|\mu(S)\|_2}$$

Cosine distance measures the difference in orientation of the two centroids while the Euclidean distance measures the absolute difference between the two centroids. Since cosine focus on angle between the vectors, it is useful in comparing the shape of data distributions rather than their scale.

**Maximum Mean Discrepancy (MMD):**  Maximum Mean Discrepancy (MMD) serves as a non-parametric two-sample test to determine if samples are drawn from the same distribution. MMD metric is identified as a particularly convenient method for assessing the similarity of real data. We followed the description in Zinat et al. Zinati et al. (2024).

**Discriminative Metric (RF AUROC):**  This metric evaluates a model's ability to distinguish between real and synthetic datasets. A random forest (RF) classifier is employed, and the area under the receiver operating characteristic (AUROC) curve is used to assess whether real and simulated cells can be effectively differentiated.

## C  ADDITIONAL RESULTS

### C.1  STATISTICAL METRICS

**Comparison of synthetically generated dataset with known causal graph approach.**  We include results for a synthetically generated dataset, LinearUniform (Table 4), to evaluate the impact of causality. Since the LinearUniform dataset was generated with a known causal graph, we applied GRNBoost2 as the causal inference method. For comparison, the non-causal model is represented in Stage 1. The results indicate that the causal model (GRNBoost2) outperforms across all three metrics.

| | Cosine distance | Euclidean distance | MMD | RF AUROC |
|---|---|---|---|---|
| *LinearUniform* | | | | |
| Control | $0.00082_{\pm 0.00012}$ | $108_{\pm 8}$ | $0.0090_{\pm 0.000}$ | $0.57_{\pm 0.016}$ |
| Stage 1 | $0.00097_{\pm 0.00010}$ | $119_{\pm 6}$ | $\mathbf{0.0139}_{\pm 0.001}$ | $0.64_{\pm 0.020}$ |
| GRNBoost2 | $\mathbf{0.00061}_{\pm 0.00011}$ | $\mathbf{93}_{\pm 9}$ | $0.0152_{\pm 0.001}$ | $\mathbf{0.63}_{\pm 0.028}$ |

Table 4: **Evaluation of synthetically generated dataset.**

### C.1.1  OVERLAPS BETWEEN HUMAN KB AND GPT-4 KB

| Dataset | # TFs | % Overlap |
|---|---|---|
| PBMC | 95 | 64.00 |
| CTL | 135 | 49.23 |
| BM | 157 | 55.82 |

Table 5: The overlaps of TFs between of Human KB and GPT-4 KB.

### C.1.2  DENSITY OF GRAPHS

| Dataset | # TFs | # Targets | # Genes | # Possible Edges | # Imposed Edges | GRN density Edges |
|---|---|---|---|---|---|---|
| $KB^H$ | | | | | | |
| PBMC | 75 | 925 | 1000 | 69375 | 9250 | 0.1333 |
| CTL | 65 | 935 | 1000 | 60775 | 9350 | 0.1538 |
| BoneMarrow | 68 | 932 | 1000 | 63376 | 9320 | 0.1471 |
| $KB^{GPT}$ | | | | | | |
| PBMC | 95 | 905 | 1000 | 85975 | 9050 | 0.1052 |
| CTL | 135 | 865 | 1000 | 116775 | 8650 | 0.0740 |
| BoneMarrow | 157 | 843 | 1000 | 132351 | 8430 | 0.0639 |
| $KB^{Llama}$ | | | | | | |
| PBMC | 266 | 843 | 1000 | 224238 | 8430 | 0.0375 |

Table 6: The density of different causal graphs wrt different KB.

### C.1.3  QUALITATIVE EVALUATION OF LLM PRIORS

The TFs that were misssing from LLM list as compared to the human curated TFs list for the PBMC data set:"PLEK", "SNAI3", "ZNF264", "ZNF708", "ZNF277","SMARCC1", "ZBTB38","BAZ2A","ARID5A","TGIF1","LYAR","DNAJC1","HOPX". These transcription factors (TFs) play diverse roles in regulating immune function, cell differentiation, and responses to stress or inflammation. Their expression in PBMCs highlights their importance in maintaining immune homeostasis, responding to infections, and their potential involvement in disease states such as cancer, autoimmune diseases, and inflammatory disorders.

Extra genes proposed as TFs by LLM: "TNFRSF18","IL2RB","NCOA4","CHD8","CDK9","LMO4", "HEXIM1","TAF10", "CCNK","POLR2A","BRD4","ANKRD11". Notably the extra transcription

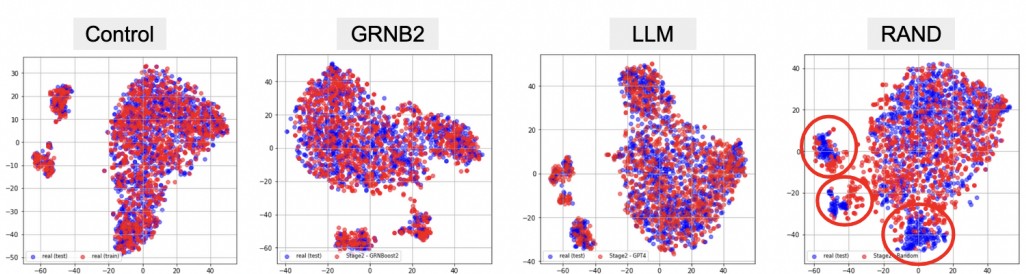

Figure 4: **TSNE projections of synthetic vs. real data for different GRN graphs (Setting 1)**. "Control" corresponds to the projection of training and testing data, "GRNB2" -the synthetic data based on GRNBoost2 graph, "LLM" - synthetic data based on LLM graph, and "RAND" - the data based on the random graph. Red dots correspond to the real data points and blue ones - to the synthetic data points. "Hallucinated" extra blue clusters in the RAND graph are marked with red circles.

factors (TFs) proposed by LLM play key roles in regulating immune cell function, influencing various aspects of immune responses, disease states (like cancer and autoimmunity), and cell differentiation pathways.

The list with the missing TFs has more genes that are not as well-defined in the literature. Specifically, genes like SNAI3, ZNF264, ZNF708, and ZNF277 have limited characterization and fewer studies addressing their specific roles in cellular processes, especially in the context of immune cells and PBMCs. In contrast, many genes in the extra proposed TFs list (e.g., TNFRSF18, CDK9, BRD4) are more extensively studied and have clearer functions, particularly in transcriptional regulation and immune processes.

C.2 QUALITATIVE EVALUATION OF SYNTHETIC DATA SETS

In addition we perform a qualitative evaluation of the synthetic data sets by visualizing them using TSNE projections (Figure 4). We observe that using random GRN induces "hallucinated" extra clusters in the data, whereas both GRNB2 and LLM proposed graphs stay faithful to the original distribution.

C.3 BIOLOGICAL EVALUATION

**Llama-KB GRNBoost2 Dataset and Improved Cell Type Segregation**    Compared to the original dataset, it was observed that the Llama-KB GRNBoost2 dataset segregates cell types more effectively. In the original dataset, four of the top five marker genes for CD4+/CD45RA+/CD25– Naïve T cells exhibited similar expression across multiple cell types, introducing noise that could complicate biological analysis. In contrast, the Llama-KB dataset demonstrated more distinct expression for CD4+/CD45RA+/CD25– Naïve T cells, with the exception of a single marker, LTB, which was expressed in other cell types as well.

**Reduction of Noise in Other Cell Types in Llama-KB GRNBoost2Dataset**    Additionally, dendritic cells in the Llama-KB GRNBoost2 model exhibited a less noisy expression pattern compared to the original dataset, a trend also observed in CD8+ cytotoxic T cells and CD4+/CD25+ T regulatory cells. A notable difference was found in CD8+/CD45RA+ cytotoxic cells, where the original dataset showed little to no expression of top markers, while the Llama-KB dataset had a more widespread, noisy expression.

**General Performance of Llama-KB GRNBoost2 in Cell-Specific Expression Profiles**    Although Llama-KB GRNBoost2 synthetic data Figure 3b introduces some noise, it generally improves cell-specific expression profiles, which could be further optimized for even cleaner cell type differentiation.

A key observation was that when mean expression was higher in specific cell types, it tended to be lower in others, with fewer than 30% of cells displaying similar expression fractions. In contrast, when expression was noisy across multiple cell types, both mean expression and cell fraction tended to be similar across these groups.

**Comparison with GPT4-KB GRNBoost2 and Random Datasets**   In comparison, the GPT4-KB GRNBoost2 dataset Figure 3a exhibited more noise than the Llama-KB dataset Figure 3b, with multiple markers expressed across various cell types at comparable mean expression levels and with higher cell fractions. The random GRN dataset resembled the Stage 1 non-causal dataset, with mean expression profiles being more clearly defined for each specific cell type. However, cell fractions for the same markers in other cell types were still significant, often exceeding 80%, indicating suboptimal gene expression specificity.

**Stage 1 Dataset and Noise Reduction in Markers**   Stage 1 data demonstrated a less noisy expression pattern, where markers were primarily confined to their specific cell types, showing lower mean expression in others. However, even though mean expression in non-specific cell types were low, cell fractions remained similar to those of the main cell type, which is not ideal for clear differentiation.

**Human-KB GRNBoost2 Dataset Performance and Noise Patterns**   For the Human-KB GRNBoost2d dataset, noise was primarily observed in a few cell types, including CD4+/CD45A+/CD25– Naïve T cells and CD8+/CD45RA+ Naïve cytotoxic T cells. Some markers from other cell types also showed similar expression across multiple cell types. These noisy patterns were not observed in the Stage 1 or random datasets and were less pronounced in the Llama-KB and GPT4-KB models, though Llama-KB GRNBoost2 outperformed GPT4-KBT in reducing noise for these specific cell types. Beyond these examples, most cell types in the Human-KB GRNBoost2 dataset did not exhibit significant noise.

**Human-KB LLM Model's Noisy Expression Profiles**   Finally, the Human-KB LLM model exhibited highly noisy or non-specific expression profiles, particularly for CD4+/CD45A+/CD25– Naïve T cells, CD8+/CD45RA+ Naïve cytotoxic T cells, CD4+ T helper cells, CD4+/CD25+ T regulatory cells, and CD4+/CD45RO+ memory cells. For other cell types, the model either failed to express most top markers or showed high cell fractions with reduced mean expression across different cell types.

**Variations in Cell Type Proportions Across Datasets**   A notable observation is that the cell type proportions in each generated dataset differ significantly from those in the original dataset, indicating a noisy overall expression profile that can alter the distribution of cell types. In the Llama-KB GRNBoost2 dataset, CD8+/CD45RA+ Naïve Cytotoxic T cells were the most abundant at 34.1%, followed closely by CD56+ NK cells, CD8+ Cytotoxic T cells, and CD4+/CD25+ T regulatory cells. This trend was similarly observed in the Stage 1, Human-KB GRNBoost2, Human-KB LLM, and GPT4-KB GRNBoost2 datasets, albeit with slightly varying percentages.

**Discrepancies in Cell Type Proportions**   The most considerable variation in proportions was noted for CD8+/CD45RA+ Naïve Cytotoxic T cells, with Llama-KB GRNBoost2 reporting the highest percentage at 34.1% and GPT-4 the lowest at 28.2%. Importantly, all generated datasets displayed significant differences when compared to the original dataset, where CD8+/CD45RA+ Naïve Cytotoxic T cells constituted only 0.1% of the population, and CD4+/CD45RA+ Naïve T cells accounted for 65.1%. These findings suggest that the models manipulate the expression profiles in such a way that the proportions of CD8+/CD45RA+ Naïve Cytotoxic T cells, CD4+/CD45RA+ Naïve T cells, and other cell types are elevated in the generated datasets.

**Overall Performance and Future Improvements for Llama-KB GRNBoost2**   Therefore, it was observed that CD4+/CD45RA+/CD25– Naïve T Cells and CD8+/CD45RA+ Naïve Cytotoxic T Cells exhibited noisy expression patterns consistently across all datasets. The original dataset performed better for CD8+/CD45RA+ Cytotoxic T Cells, showing lower noise in their expression compared to the Llama-KB GRNBoost2 generated dataset. Among the various models, the LLAMA model emerged as the best performer, providing clearer segregation of cell types and reduced noise, particularly for CD4+/CD45RA+/CD25– Naïve T Cells and dendritic cells.

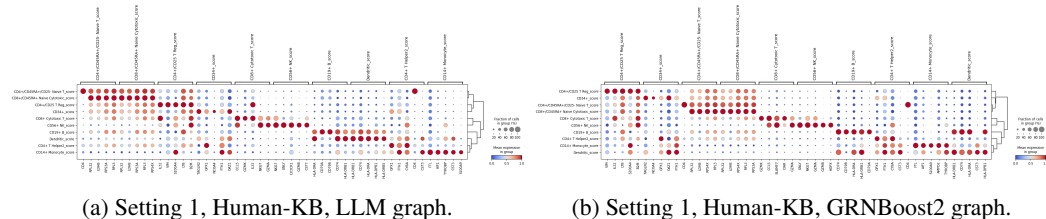

(a) Setting 1, Human-KB, LLM graph.          (b) Setting 1, Human-KB, GRNBoost2 graph.

Figure 5: Gene expression analysis. Human-KB LLM and Human-KB GRNBoost2 datasets.

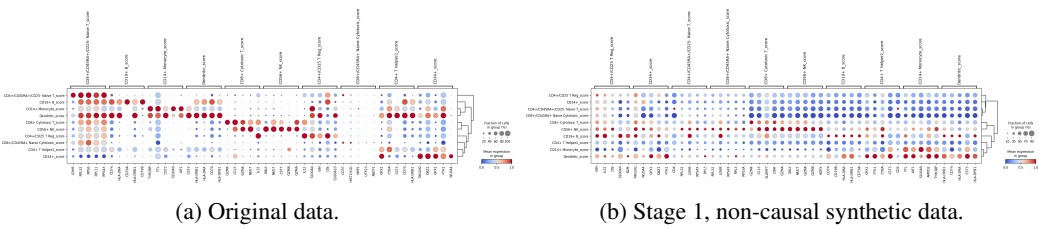

(a) Original data.          (b) Stage 1, non-causal synthetic data.

Figure 6: Gene expression analysis. Original and Stage 1 datasets.

**Conclusion: Challenges and Future Refinements**   In contrast, the GPT4-KB GRNBoost2 model exhibited more noise than Llama-KB GRNBoost2, with similar markers expressed across various cell types at comparable mean expression levels. The Stage 1 model displayed less noisy expression patterns, but some markers were still expressed significantly across non-specific cell types, while the random-generated dataset had poorer cell-specific profiling. In conclusion, the discrepancies in cell-type proportions between generated datasets and the original dataset highlight the challenges in achieving accurate cell-type specificity. These variations can significantly impact downstream analyses and interpretations in biological research. Consequently, refining these models to enhance the specificity of marker expression is essential for producing more reliable datasets that reflect the true cellular composition observed in biological samples.

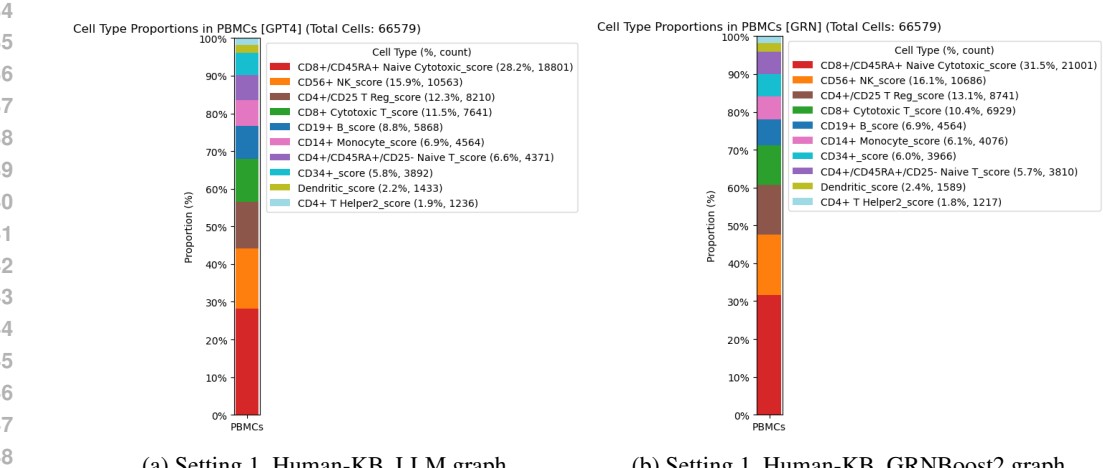

(a) Setting 1, Human-KB, LLM graph      (b) Setting 1, Human-KB, GRNBoost2 graph

Figure 7: Cell annotation and proportions analysis. Human-KB LLM and Human-KB GRNBoost2 datasets.

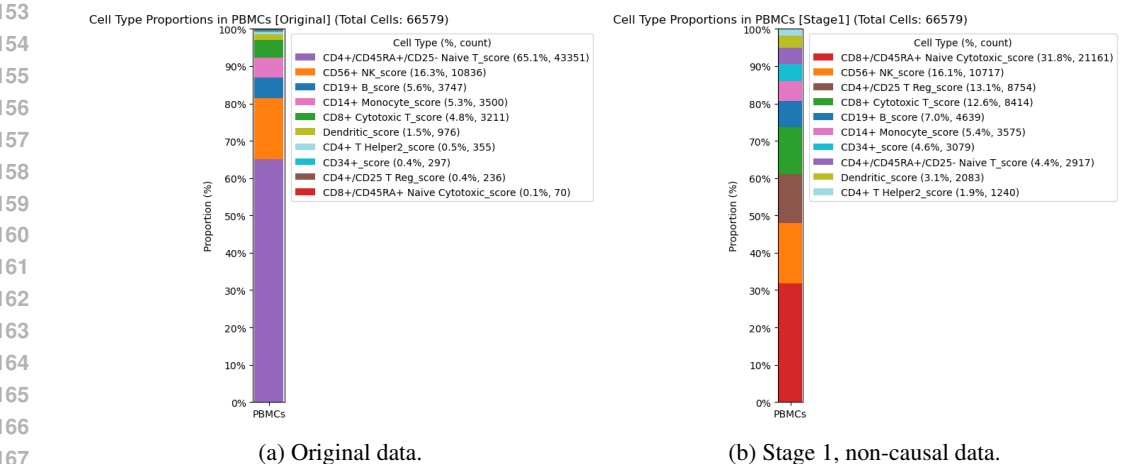

(a) Original data.      (b) Stage 1, non-causal data.

Figure 8: Cell annotation and proportions analysis. Original and Stage 1 datasets.

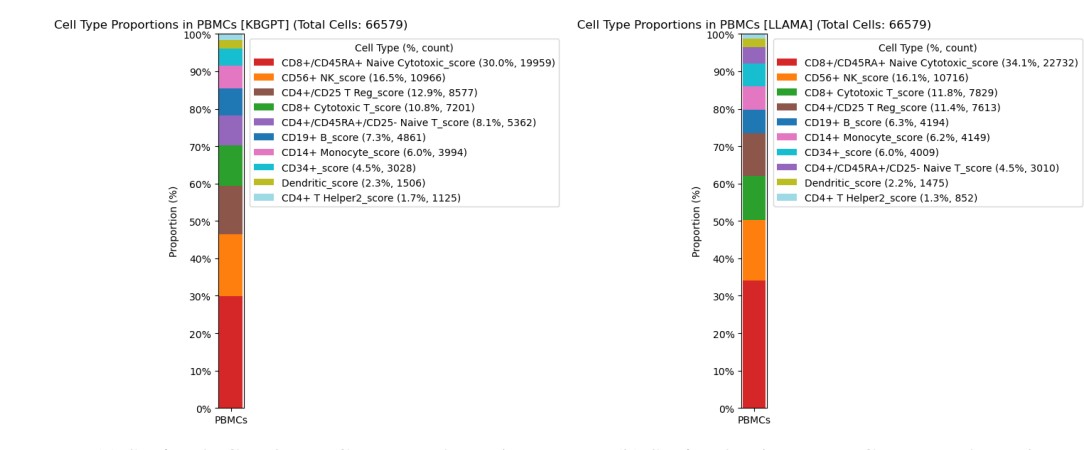

(a) Setting 2, GPT4-KB, GRNBoost2 graph.      (b) Setting 2, Llama-KB, GRNBoost2 graph.

Figure 9: Cell annotation and proportions analysis. GPT4-KB LGRNBoost2 and Llama-KB GRNBoost2 datasets.

