# OpenReview forum: "LLM4GRN: Discovering Causal Gene Regulatory Networks with LLMs - Evaluation through Synthetic Data Generation"
_ICLR.cc/2025/Conference — Submitted to ICLR 2025_

### Official Review · Reviewer_ftKG · 2024-10-28

**Soundness:** 2
**Presentation:** 2
**Contribution:** 2
**Rating:** 3
**Confidence:** 4

**Summary:**

The paper titled "LLM4GRN: Discovering Causal Gene Regulatory Networks with LLMs – Evaluation through Synthetic Data Generation" presents a novel approach leveraging large language models (LLMs) for inferring gene regulatory networks (GRNs) from single-cell RNA sequencing (scRNA-seq) data. The authors propose using LLMs to generate complete GRNs or to provide prior knowledge to traditional statistical methods. A key contribution is the use of causal synthetic data generation as an evaluation strategy, which allows for the assessment of the inferred GRNs in the absence of ground truth causal graphs. The study demonstrates the potential of LLMs in supporting statistical modeling and data synthesis for biological research, with a particular emphasis on the combination of LLMs and statistical inference methods showing promise for scRNA-seq data analysis.

**Strengths:**

1. The methodology involving the use of causal synthetic data generation to evaluate GRNs is well-designed and addresses a significant challenge in the field, namely the lack of ground truth data for validation.
2. The paper is well-organized, with clear explanations of the methodology, experimental setup, and results. The use of visual aids such as figures and tables enhances the understanding of the presented material.
3. The potential of LLMs to capture complex biological interactions and support statistical modeling in genomics is a significant contribution, with broad implications for disease mechanism understanding and therapeutic target identification.

**Weaknesses:**

1. The paper could benefit from a more detailed discussion on the limitations of the current approach, particularly regarding the assumptions made in the GRN structure due to the constraints of the GRouNdGAN framework.
2. While the study highlights the potential of LLMs, it does not explore the reasons behind the performance differences observed between the LLM-based and statistical methods, which could provide further insights into the underlying mechanisms.
3. The paper might consider discussing the scalability of the proposed method, especially how it might perform with larger and more complex datasets.
4.The paper does not seem to have much theoretical innovation. It simply uses LLM as a plug-in to replace previous human prior knowledge or statistical analysis methods.
5. The results are not very convincing: Figure 2 aims to illustrate the robustness and superiority of the method by overleaping different methods. However, the small gap cannot reflect the advantages of the proposed method.

**Questions:**

1. Could the authors elaborate on how the performance of the LLM-based GRN inference might be affected by the quality and representativeness of the data used to train the LLMs?
2. It would be helpful if the authors could discuss potential future work on refining the GRN inference process, especially in addressing the noise and specificity issues mentioned in the biological plausibility analysis.
3. The paper mentions the use of GPT-4 and Llama-3.1-70B models. Are there any plans to compare the approach with other LLMs or to investigate the impact of model size on the performance of GRN inference?

---

> ### Author Response · Authors · 2024-11-22
> **Answers to the Reviewer ftKG**
>
> We are grateful to reviewer ftKG for their detailed review of our work.
>
> > Question 1: Could the authors elaborate on how the performance of the LLM-based GRN inference might be affected by the quality and representativeness of the data used to train the LLMs?
>
> We thank the reviewer for the relevant observation and point to the "Discussion and Limitations" section. We have also included the relevant information from the  "Discussion and Limitations" to answer Q5 to the second reviewer, who raised a similar question.
>
> > Question 2: It would be helpful if the authors could discuss potential future work on refining the GRN inference process, especially in addressing the noise and specificity issues mentioned in the biological plausibility analysis.
>
> We appreciate the reviewer’s suggestion and agree that further refinements are needed.W However, we plan to generate cell type-specific TFs to cover all important TFs for each cell type in each dataset and generate more biological driven GRNs.One of the directions we envision is learning causal graphs for each cell type separately, as the graphs can be  cell-type specific. **However, we emphasize the biological plausibility of our proposed method for the PBMC data set is better than the state of the art**.
>
> > Question 3: The paper mentions the use of GPT-4 and Llama-3.1-70B models. Are there any plans to compare the approach with other LLMs or to investigate the impact of model size on the performance of GRN inference?
>
> We appreciate the suggestion. Our results demonstrate strong performance with open-source model, which encourages further exploration of even smaller models.
> We used Llama-3.1-8b to generate potential TFs and observed that there existed 82.7% overlap for PBMC, 94.2%overlap for BM, and 77.6% overlap for BM. Additionally compared to Llama-3.1-70b model, we observed Llama-3.1-8b struggled with instruction following. In contrast, Llama-3.1-70B demonstrated robust performance while maintaining a good balance between computational cost and accuracy, making it a favorable choice for our GRN inference tasks.
>
>
> We hope our reply clarifies the reviewer's concerns and would be happy to answer any further question the reviewer may have.

---

> > ### Comment · Reviewer_ftKG · 2024-11-26
> >
> > Thank you for your response. However, we find it difficult to consider this article sufficiently innovative. We will maintain the current score.

---

> > > ### Author Response · Authors · 2024-11-27
> > >
> > > We thank the reviewer for their comment, however we respectfully disagree with the assessment regarding the level of innovation. To the best of our knowledge, we are the first to leverage domain knowledge via LLMs to enhance causal generative models with GRNs. We expect our approach, which integrates domain and context-specific insights through LLMs, to enable followup research not only in the ML community but also in the intersection of LLM and bio/health domains.

---

### Official Review · Reviewer_h4j7 · 2024-11-01

**Soundness:** 2
**Presentation:** 3
**Contribution:** 2
**Rating:** 5
**Confidence:** 4

**Summary:**

This paper presents LLM4GRN, a method that integrates Large Language Models (LLMs) to discover causal Gene Regulatory Networks (GRNs) from single-cell RNA sequencing (scRNA-seq) data. The study employs a causal synthetic data generation approach, comparing two experimental setups where LLMs either generate GRNs directly or assist statistical methods (e.g., GRNBoost2) by providing transcription factor (TF) candidates. The authors test the approach on three datasets, highlighting LLMs' effectiveness in biological plausibility and similarity to real data. Notably, Llama-3.1 outperformed GPT-4 in some aspects, suggesting potential advantages for using certain open-source models over more advanced commercial ones.

**Strengths:**

1. Innovative Use of LLMs: Leveraging LLMs in GRN discovery demonstrates novel application potential in biological data analysis.
2. Comparison of Multiple LLMs: Provides comparative insights into GPT-4 and Llama-3.1 performances, showing valuable differences that inform model selection for scRNA-seq analysis.
3. Promising Hybrid Approach: Results show that combining LLM-derived TF lists with statistical models like GRNBoost2 can improve causal inference quality.

**Weaknesses:**

1. Limited Dataset Variety: Only three datasets (PBMC-All, PBMC-CTL, and Bone Marrow) are used, which limits generalizability to broader biological contexts.
2. Dependency on Pathway Completeness: The method’s performance may suffer in cases with incomplete pathway information, which isn’t fully addressed.
3. Biological Relevance of Generated Data: Some results indicate noise and inconsistency in cell-type differentiation, which could impact downstream analyses and applications.
4. Complexity of Methodology: The use of multiple knowledge bases and LLM models creates a complex pipeline that may be difficult for other researchers to reproduce.

**Questions:**

1. How might the LLM-based approach handle incomplete pathway information or limited TF-gene knowledge in certain datasets?
2. How does pathway information specifically improve GRN inference compared to simpler approaches that omit it?
3. What criteria were used to select the three datasets, and how might the results generalize to datasets with different biological characteristics?
4. How does the complexity of LLM4GRN impact reproducibility, and are there ways to simplify the model without losing performance?
5. Can you discuss how robust the model is to potential biases in the LLM-based knowledge base, especially given the dependency on prior information?
6. Why do you believe Llama performs better than GPT-4 in certain settings, and what does this imply for LLM-based GRN inference?

---

> ### Author Response · Authors · 2024-11-22
> **Answers to the Reviewer h4j7**
>
> We appreciate reviewer h4j7’s detailed review of our work. We provide in-line responses to your questions below:
>
> > Question 1&2:
> > - How might the LLM-based approach handle incomplete pathway information or limited TF-gene knowledge in certain datasets?
> > - How does pathway information specifically improve GRN inference compared to simpler approaches that omit it?
>
> We appreciate the reviewer’s question and would first like to clarify our understanding of “pathway information” to ensure we address the concerns accurately. We suppose the reviewer is referring to curated biological knowledge, such as well-established interactions between transcription factors (TFs) and target genes.
> In our approach, pathway information enhances GRN inference by providing biological context that guides the statistical algorithm toward more context-specific and biologically relevant network predictions. Our results demonstrate that this approach produces synthetic data that is not only more faithful to biological constrains but also more biologically plausible compared to methods that omit it (see below figure for Stage 1 non-causal synthetic dataset comparison).
>  **Moreover, unlike traditional methods that depend on manually curated, static knowledge bases -- a process that is both labor-intensive and prone to human error -- our LLM prior-based approach dynamically integrates pathway data by consulting multiple knowledge sources**.
>
> Additionally, we propose a hybrid strategy that leverages LLM to gather information about potential TFs and combines this with data-driven statistical inference to address gaps in pathway information.
>
> > Weakness 1 & Question 3:
> > - Limited Dataset Variety: Only three datasets (PBMC-All, PBMC-CTL, and Bone Marrow) are used, which limits generalizability to broader biological contexts.
> > - What criteria were used to select the three datasets, and how might the results generalize to datasets with different biological characteristics?
>
> **We would like to clarify that while our study focuses on three datasets, they were chosen to represent sufficient diversity across key biological contexts**. Specifically, the PBMC-ALL dataset is a human dataset, the Bone Marrow dataset represents mouse data, and PBMC-CTL focuses on a specific cell type. This selection allows us to capture variations across species (human vs. mouse) as well as across different biological and cellular granularity, providing meaningful insights and demonstrating the versatility of our approach. **Additionally, the datasets were selected to enable direct comparison with the GRouNdGAN paper since we considered the approach proposed by Zinati et al. as a baseline (current state of the art).**
>
> > Question 4: How does the complexity of LLM4GRN impact reproducibility, and are there ways to simplify the model without losing performance?
>
> To maintain simplicity and reproducibility, we make sure that all the experiments have the fixed seed, the code is packaged and well documented. We described the experimental setup is described in detail in Appendix A. We also deliberately evaluated Llama so it can be run locally without the need for external cost associated with Open-AI GPT-4 models to ensure full reproducibility.
> Additionally, we utilize a straightforward process where the LLM is prompted to generate a list of transcription factors (TFs), which are then seamlessly incorporated into the GRouNdGAN workflow and will release the code after publication.  We also deliberately evaluated Llama so it can be run locally without the need for external cost associated with Open-AI GPT-4 models to ensure full reproducibility.
>
>
> > Question 5: Can you discuss how robust the model is to potential biases in the LLM-based knowledge base, especially given the dependency on prior information?
>
> We agree that the LLMs can be prone to biases in the data and have provided the discussion in the paper. Due to the lack of space, we point the reviewer to the Section 5.4 “Discussion And Limitations”.
>
> > Question 6: Why do you believe Llama performs better than GPT-4 in certain settings, and what does this imply for LLM-based GRN inference?
>
> We thank the reviewer for raising this question. **To address this point, we conducted an additional ablation study to investigate differences between Llama and GPT-4 (It can now be accessed in the supplementary materials). Our findings indicate that Llama’s superior performance in certain settings is primarily due to its ability to propose a more comprehensive initial list of transcription factors (TFs)**. This larger TF list provides the statistical algorithm with a broader foundation, enabling it to better infer relevant relationships from the data. However, we also note that Llama underperforms in settings where the statistical algorithm is not included in the pipeline. This suggests that Llam’s strength lies in its capacity to complement the statistical inference process, rather than functioning as a standalone tool.

---

> > ### Author Response · Authors · 2024-11-28
> >
> > We thank again Reviewer h4j7 and hope we have answered all the questions.
> > Please let us know if any further clarifications are needed.

---

> > > ### Comment · Reviewer_h4j7 · 2024-12-02
> > >
> > > Thank you for your response. While I acknowledge the potential value of using LLM for GRN construction for the community, the current version of this paper still has some gaps. I believe additional benchmarking is necessary to substantiate the findings. Therefore, I will stick with the same score for now.

---

### Official Review · Reviewer_HajQ · 2024-11-05

**Soundness:** 2
**Presentation:** 1
**Contribution:** 2
**Rating:** 5
**Confidence:** 4

**Summary:**

This paper explores the application of general-purpose foundation models (GPT-4, LLaMA) for identifying context-specific transcription factors (TFs) and constructing corresponding gene regulatory networks (GRNs). The authors evaluate the GRNs by contrasting them with knowledge base-derived GRNs and those inferred using GRNBoost2, then apply causal synthetic data generation based on the method of Zinati et al. (2024) to assess the biological plausibility of the generated networks.

**Strengths:**

- The study addresses a critical need for context-specific GRNs, as many practitioners rely on generic knowledge-base GRNs, which lack context specificity and often lead to less accurate inferences.
- A thorough analysis, evaluating both the TF lists generated by foundation models and the resulting GRNs with controls at each step

**Weaknesses:**

The primary limitation is the lack of objective metrics for validating the inferred GRNs. While the authors use synthetic data generation as an indirect evaluation method, results vary significantly across datasets, with the GPT-based GRNs performing well on PBMC_ALL but showing weaker, near-random performance on other datasets (as measured by MMD and RF AUROC). These results indicate low systematic performance on the author's suggested standards.

The above may also be a result of knock-effects caused by the Causal GAN synthetic data generation. To address these issues, it would be helpful to incorporate additional validation methods that test the sensitivity of the probing strategy to known, context-dependent variations within well-studied GRNs. For instance, starting with a validated GRN and introducing experimentally confirmed context-specific variations could help determine if the models correctly adapt the network structure. Although this approach still has limitations (e.g., it could have been in the training set of GPT4), it could provide orthogonal evidence of the probing strategy’s effectiveness in identifying/retrieving context-sensitive elements.

Similarly, for the initial TF lists generated by foundation models, I recommend testing their context specificity by comparing TF lists for pairs of contexts with low TF overlap based on expression in resources like the Human Cell Atlas. This could reveal whether the models distinguish between TFs active in distinct biological contexts.

**Questions:**

- Please consider adding other orthogonal direct ways of validating GRNs, such as the one suggested above by testing the exact same probing strategy on whether it can identify context specific variations of a well known and validated TF GRN
- Please include an orthogonal method to validate the context-specific TF list, such as distinguishing TFs based on uncorrelated expression patterns across distinct cell types, as mentioned in the Weaknesses section

---

> ### Author Response · Authors · 2024-11-22
> **Answers to the Reviewer HajQ**
>
> We thank reviewer HajQ for their detailed review of our work.
>
> > Question 1&2:
> > - Please consider adding other orthogonal direct ways of validating GRNs, such as the one suggested above by testing the exact same probing strategy on whether it can identify context specific variations of a well known and validated TF GRN
> > - Please include an orthogonal method to validate the context-specific TF list, such as distinguishing TFs based on uncorrelated expression patterns across distinct cell types, as mentioned in the Weaknesses section.
>
> We appreciate the reviewer’s insightful suggestion to include additional orthogonal validation strategies for GRNs, such as testing our probing on context-specific variations of a well-known and validated transcription factor GRN. However, we acknowledge that the lack of reliable ground truth GRNs presents a significant challenge in such validation efforts. **Our approach is specifically designed to address this limitation by focusing on practical utility and demonstrating robust performance in the absence of established ground truth.** Furthermore, our results are supported by both statistical and biological evaluations. They demonstrate the effectiveness of our method, combining GPT-4 knowledge base with a statistical GRN inference approach, compared to the baseline.
>
>
> > Weakness 1: The primary limitation is the lack of objective metrics for validating the inferred GRNs. While the authors use synthetic data generation as an indirect evaluation method, results vary significantly across datasets, with the GPT-based GRNs performing well on PBMC_ALL but showing weaker, near-random performance on other datasets (as measured by MMD and RF AUROC). These results indicate low systematic performance on the author's suggested standards.
>
> Thank you for pointing out this concern. We appreciate the opportunity to clarify. **We would like to highlight that the performance of our approach remains consistent across most datasets analyzed. Specifically, both PBMC-ALL and PBMC-CTL demonstrate that GPT+GRNB2 is the top-performing method.** While there is a minor variation in performance for the Bone Marrow dataset—specifically in terms of Cosine, Euclidean, and MMD distances between Human+GRNB2 and GPT+GRNB2—these differences are not substantial enough to indicate a systemic issue. For example, when examining the Euclidean distance score, Human+GRNB2 achieved 7+- 5 compared to 78 +- 4 for GPT+GRNB2, demonstrating a marginal 1-point difference in favor of the existing method. In contrast, for the PBMC-CTL dataset, GPT+GRNB2 outperformed with a more notable margin, achieving 57 +- 6 versus 65 +-6 for Human+GRNB2, reflecting an 8-point difference in our method’s favor. Additionally, as noted in the paper (Line 484), the observed lower performance of large language models (LLMs) on mouse data aligns with the limited availability of encoded information for such data, a factor that we discussed. We hope this context helps clarify the observed results and highlights the robustness of our approach across different datasets.
>
> We hope our reply clarifies the reviewer's concerns and would be happy to answer any further question the reviewer may have.

---

> > ### Comment · Reviewer_HajQ · 2024-11-27
> >
> > Thank you for your responses. The issues that I raised have as general theme the lack of translation from the in-silico to the biology domain, which makes it very hard to asses the validity of the method. The benchmarking so far I don't believe gets us there. I do believe it is reasonable to have as an expectation that a method like this is able to have good performance on validated GRNs that have had substantial amounts of experimental backing: like p53 and the JAK/STAT pathways. I will maintain the current score.

---

> > > ### Author Response · Authors · 2024-11-27
> > >
> > > We thank the reviewer for bringing up the in-silico to the biology translation perspective, and agree that it can be a valuable direction for future research; However, it is not the primary focus of this submission. While we recognize that the paper may attract interest from various communities (bioinformatics, health, and ML), our current focus is on the ML contribution, specifically on synthetic data generation. Synthetic data is a highly active topic within the ML community, and we believe our work brings a relevant contribution in improving state of the art in synthetic data generation by incorporating LLM in the pipeline.

---

### Author Response · Authors · 2024-11-22
**General Response**

We thank all the reviewers for their valuable feedback and thoughtful comments. We are pleased to hear that the reviewers appreciated our **innovative use of LLMs for GRN networks which addresses a need for context specific biological interactions** (Reviewers HajQ, h4j7 & ftKG). We are also encouraged by Reviewer ftKG's acknowledgment of the **novel causal graph evaluation framework on synthetic data, which addresses a significant challenge in the field**.

We are equally grateful for the suggestions to include more data sets (Reviewers h4j7 & ftKG), consider more complex GRN structures (Reviewers h4j7 & ftKG) and more traditional, ground truth graph based evaluation approaches (Reviewer h4j7). We plan to expand our study in these directions in the future. However, as outlined in the paper, our current work is set in the context of GroundGAN, the current state-of-the-art algorithm for synthetic ScRNA seq data generation. For this study, we adhered to the GroundGan pipeline for synthetic data generation and used it as a baseline (without LLMs). **Importantly, our enhanced approach, which integrates LLMs into the pipeline, demonstrates clear and consistent improvements over the baseline, as detailed in our results.** We also note that datasets utilized in our experiments are diverse, encompassing both animal and human data, as well as varying in cellular context, including experiments involving multiple cell types and single cell types.

Below, we address the individual questions the reviewers asked. We hope our reply clarifies the reviewer's concerns and would be happy to answer any further question the reviewer may have.

---

### Meta-Review · Area_Chair_EEXa · 2024-12-20

**Metareview:**

This paper investigates leveraging LLMs to discover Gene Regulatory Network (GRN) from scRNA-seq data, evaluated via synthetic data generation. While reviewers appreciated the novelty and potential impact, concerns were raised about limited datasets, lack of robustness to noise and biases, minimal benchmarking against validated GRNs, and marginal performance improvements. The authors addressed some concerns by clarifying the theoretical basis, adding ablation studies, and outlining future directions, but many reviewers found these responses insufficient. Overall, the paper presents an interesting idea but requires further evaluation and refinement.

**Additional Comments On Reviewer Discussion:**

All reviewers acknowledged the rebuttal. The reviewers recognized the potential value of using LLMs for GRN construction within the community. However, they noted that the current version of the paper still contains significant gaps.

---

### Decision · Program_Chairs · 2025-01-22

Reject